# A Qualitative Study on Barriers to COVID-19 Vaccine Uptake among Community Members in Tanzania

**DOI:** 10.3390/vaccines11081366

**Published:** 2023-08-15

**Authors:** Melina Bernard Mgongo, Rachel N. Manongi, Innocent B. Mboya, James S. Ngocho, Caroline Amour, Monica Mtei, Julieth S. Bilakwate, Ahmed Yusuph Nyaki, Johnston M. George, Beatrice J. Leyaro, Amina Farah, James T. Kengia, Florian Tinuga, Abdalla H. Bakari, Fatimata B. Kirakoya, Awet Araya, Ntuli A. Kapologwe, Sia E. Msuya

**Affiliations:** 1Institute of Public Health, Kilimanjaro Christian Medical University College (KCMUCo), Moshi P.O. Box 2240, Tanzania; rachel.manongi@kcmuco.ac.tz (R.N.M.); innocent.mboya@med.lu.se (I.B.M.); james.ngocho@kcmuco.ac.tz (J.S.N.); caroline.amour@kcmuco.ac.tz (C.A.); monica.mtei@kcmuco.ac.tz (M.M.); julietsebba@gmail.com (J.S.B.); ahmed.nyaki@kcmuco.ac.tz (A.Y.N.); johnston.george@kcmuco.ac.tz (J.M.G.); beatrice.john@kcmuco.ac.tz (B.J.L.); siaemmanuelimsuya@gmail.com (S.E.M.); 2Department of Translational Medicine, Lund University, 214 28 Malmo, Sweden; 3Faculty of Epidemiology and Population Health, London School of Hygiene and Tropical Medicine, London WCIE 7HT, UK; 4Joint Malaria Program, Kilimanjaro Christian Medical Centre, Moshi P.O. Box 3010, Tanzania; 5President’s Office—Regional Administration and Local Government, Dodoma P.O. Box 1923, Tanzania; james.tumaini@tamisemi.go.tz (J.T.K.); ntuli.kapologwe@tamisemi.go.tz (N.A.K.); 6Immunization and Vaccine Development Program, Ministry of Health, Dodoma P.O. Box 743, Tanzania; florian.tinuga@afya.go.tz; 7School of Natural Science, The State University of Zanzibar, Tunguu P.O. Box 146, Tanzania; abdalla.bakar@suza.ac.tz; 8United Nations Children Fund (UNICEF), Dar es Salaam P.O. Box 4076, Tanzania; fbalandi@unicef.org (F.B.K.); aaraya@unicef.org (A.A.); 9Community Health Department, KCMC Hospital, Moshi P.O. Box 3010, Tanzania

**Keywords:** COVID-19, COVID-19 vaccines, vaccine hesitancy, vaccine uptake, barriers, facilitators, perceptions, qualitative, Tanzania

## Abstract

The use of vaccines is one of the key tools in reversing the COVID-19 pandemic; however, various reports reported the low uptake of the vaccines. This study explored the barriers to the COVID-19 vaccine uptake among community members in Tanzania. A qualitative explorative study was conducted in December 2021 and April 2022 in eight regions of Tanzania. Focus group discussions (FGDs) and in-depth interviews (IDIs) were the methods of data collection. A total of 48 FGDs and 32 IDIs were conducted. Participants were aware of the COVID-19 disease and vaccines. The barriers to the COVID-19 vaccine non-uptake included receiving contradicting statements from top government leaders, vaccine preceded the education, myths towards vaccines, the presence of different types of vaccines, the process of getting the vaccine, the influence of social media and random people from the community, and vaccine conflicting religious beliefs. Despite being aware of the vaccine, the uptake of the COVID-19 vaccine is still low. Interventions that focus on increasing community knowledge about COVID-19 vaccines and addressing myths about the vaccines are needed.

## 1. Introduction

Vaccine hesitancy refers to the delay in acceptance or refusal of vaccination despite the availability of vaccination services [1]. Vaccine hesitancy is listed among the 10 health global threats that affect the globe [2]. Vaccine hesitancy prevents global progress in tackling all vaccine-preventable diseases due to its influence on vaccine uptake [2,3]. Previous studies have identified and documented various factors that relate to vaccine hesitancy. These included religion and culture [4,5], perceived risk or benefits of receiving the vaccine, perceived importance of the vaccine, fear of side effects, lack of vaccine confidence, geographical accessibility, cost, mode of administration, the recommendations from health care providers, lack of information, and media or communication [4,5,6,7,8]. 

The coronavirus disease 2019 (COVID-19), a threat to public health, was declared a global pandemic by the WHO in March 2020. Evidence from cohort and randomized clinical trials have shown that the COVID-19 vaccine reduces the overall infection rate, hospitalization, severe infections, and deaths [9,10,11,12]. Despite the evidence, COVID-19 vaccine uptake has been low in many Sub-Saharan Africa (SSA) countries, including Tanzania [8,13]. In Tanzania, there are also disparities in the uptake of the COVID-19 vaccine, with Dar es Salaam having the lowest rate of 8% and Simiyu having 38% [8]. Studies have shown suboptimal knowledge of COVID-19 vaccines, misconceptions, concerns about vaccine safety and side effects, lack of trust in pharmaceutical industries, misinformation or conflicting information from the media, fear of negative effects, risk perceptions, and lack of confidence in the new vaccine as factors leading to hesitancy, and ultimately affects the COVID-19 vaccine uptake [4,7,14]. 

Tanzania was late in introducing the COVID-19 vaccines compared to other SSA countries due to inadequate political will and leadership choices [15,16]. Studies have reported disparities in vaccine acceptance rates in SSA countries. A systematic review conducted in six countries of SSA showed that vaccine acceptance was almost universal in Ethiopia, whereby 98% were willing to be vaccinated [14]. A study in Kilimanjaro Tanzania revealed that 65% of the population had negative attitudes towards COVID 19 vaccines and 28% of those interviewed were reluctant to be vaccinated due to various reasons such as side effects and unknown safety [17]. In 2021, history changed, and the president of the United Republic of Tanzania, H.E Samia Suluhu Hassan, officially introduced the vaccines, and she was vaccinated publically [16]. The first vaccines were thus introduced in June 2021 in an effort to halt the pandemic. The vaccines are given voluntarily to any person aged 18 and above. There are five COVID-19 vaccines that are available in the country (Janssen, Pfizer, Moderna, Sinopharm, and Sinovac) and have been approved by the WHO. In the first phase, the country had a target to vaccinate 20% (11 million) of the population and, in December 2022, to vaccinate 60% of the eligible population. However, by October 2022, data from the Ministry of Health of Tanzania (MoH) showed that only 44% of the eligible population were fully vaccinated, with limited data on community perceptions of the COVID-19 vaccines and their reasons for hesitancy. Further, the national vaccination data, at the beginning of vaccination, showed a lower rate of vaccination among women than men, possibly because of fear, lack of confidence in the effectiveness of vaccines, and fear of infertility, among other reasons shown elsewhere [14,18]. 

Tanzania joined the global partners to push the COVID-19 vaccination targets to reach the target by December 2022 [19]. The Minister of Health, UN agencies, and local partners launched a massive vaccination campaign to create awareness and demand, with a target to vaccinate 4 million people across all 26 regions of mainland Tanzania. In-depth knowledge of community knowledge and perceptions on the COVID-19 vaccine is important to guide health promotion campaigns by different stakeholders. The study was therefore conducted to understand the community knowledge, perceptions, attitudes, and practices, which are the main influencers of behavior. The study used qualitative methodology to explore the physical, sociocultural, and environmental barriers influencing vaccine acceptance. 

## 2. Materials and Methods

A community-based cross-sectional study using a mixed-methods study design was conducted from December 2021 to April 2022. The study was conducted in six regions of Tanzania’s mainland and two regions in Zanzibar. The regions were purposively selected based on the vaccine wastage rate of September 2021. Vaccine wastage was defined as the number of doses which were opened and discarded due to a lack of people to vaccinate. The selected regions were Dar es Salaam, Lindi, Kilimanjaro, Tabora, Simiyu, and Mbeya in the Tanzania mainland and Mjini Magharibi and Pemba Kaskazini in Zanzibar. In each region, two districts were purposively selected, one representing an urban and another representing a rural district, providing a total of 16 districts that participated. At the district level, two to six wards or *shehia* (smallest government administrative unit in Pemba and Zanzibar) were randomly selected. The details sampling methods, sample size calculation, and statistical analysis of the quantitative study are reported elsewhere [6,8]. This paper focuses on qualitative data. 

The study population included participants aged 18 years or above who consented to participate. In-depth interviews (IDIs) and focus group discussions (FGDs) were the methods used to collect the data. The IDIs included two religious leaders, one political leader who may be a ward councilor, a ward/village leader, a teacher, and/or another influential person. In each district, three FGDs were conducted; one was with women who were vaccinated; another with women who were not vaccinated; and a third FGD was with men aged 18 years and above. The three groups were chosen since there was a higher vaccine hesitancy among women than men. By October 2021, the vaccine uptake among men was 65% vs. 35% of women, and there was a need to explore and understand the perceptions of COVID-19 and reasons for hesitancy in separate groups among women. The total number of people in each FGD ranged from 8–12, as this was dependent on the availability of participants. 

On the day of data collection, the research team met at the ward leader offices, where in most cases, the street leaders and/or other selected link persons came to pick the study team. The street leaders/ link person introduced the researchers to the study participants. Before the IDIs, all the participants were informed about the study and its aims. For those agreeing to participate, they verbally consented and were asked for permission to record the interviews. Most of the IDIs were held at ward or street leaders’ offices and took 45 to 60 min to be completed. The FGDs were conducted in selected quiet areas. Most of the FGDs were conducted at ward or village leaders’ offices, and others were conducted at the dispensaries depending on the preferences of the participants. In the FGDs, the members were also asked for consent and for permission to record the interviews. Three members facilitated the FGDs, and the interviews lasted for about 60 min. The aim of both IDIs and FGDs was to explore in-depth the aspects covered in the household survey, including the general knowledge and attitude towards COVID-19 and vaccines and the related cultural beliefs and taboos, and power dynamics. Factors affecting the uptake of vaccines were explored in detail, including the perceived risks and trust towards COVID-19 vaccines. FGDs were conducted with a group of women who have reported having taken a vaccine and a group that has reported not having taken the vaccine to understand their general reflections related to COVID-19 and vaccines and the reasons for their decisions.

The quality assurance and quality control of data collection were conducted on a daily basis by research assistants under the mentorship of the senior researchers. Daily debriefings were done to discuss the interviews and emerging issues that need to be incorporated into the next interviews. The senior team member at each region had to write a daily summary of the interview(s) conducted following an agreed format and send it to the leader of the qualitative team. The recorded interviews were uploaded into the secure folder created and managed by the qualitative team leaders. The recorded interviews were transcribed verbatim and then translated into English. Coding and then thematic framework analysis was used to summarize the data.

### Data Analysis

Thematic framework data analysis was performed as per Braun and Clarke (2006). The steps involved the familiarization of data whereby the in-depth and FGD scripts were transcribed verbatim, read, and translated to English by competent Swahili and English speakers. This was then followed by the development of initial codes by two independent coders (MM, BJL) using the ATLAS.ti program. The prior codes were developed from the IDI and FGD guides and grounded emergent codes were developed from the raw data in the transcripts. The codes were identified through the examination of raw data and by assigning words or phrases that captured the meaning of the data. The after codes were organized and categorized so that they can be interpreted and compared by the authors. In case there was a disagreement, another coder (RNM) joined the team to solve it. The authors agreed on the developed codes, and for any newly emerged code, the coders discussed it, and after the agreement, the codes were added to the code list. The next step involved searching for themes, and all codes that were related were sorted and listed into one comprehensive theme. The coders used the matrix table to list all the codes that were related, sorted, and listed in one theme. The themes were developed to answer the research question, and the data extracts were organized to each theme. After theme development, the last step was reviewing and refining themes and report writing.

## 3. Results

### 3.1. Sociodemographic Characteristics

A total of 48 focus group discussions (FGDs) and 32 in-depth interviews (IDIs) were conducted. Three hundred and sixty-nine participants participated in the FGDs (Table 1). The mean age of the participants in the FGDs was 40 years, with a range of (18-94). A total of 69% of the participants were females, 65% were married, 51% were self-employed, and 13% were engaged in farming.

A total of 32 participants participated in the IDIs, as seen in Table 2. Their age ranged from 28–74 years, with 91% being married, 67% with secondary education or higher, and 45% with employment.

### 3.2. Awareness and Knowledge about the COVID-19 Infection

The majority of the participants in the IDIs and FGDs were aware of the COVID-19 virus. The following are sub-themes that explain their awareness of COVID-19.

#### 3.2.1. COVID-19 Discovery and Mode of Transmission 

Across all FGDs and IDIs, participants were aware and knowledgeable of the COVID-19 discovery and mode of transmission. The majority declared that COVID-19 had hit the world since its discovery. Various stakeholders explained that “COVID-19 is a disease that originated in China in 2019, spread with a virus that is believed to come from animals” (IDI 2-Pemba). The participants further mentioned that “disease affects the respiratory system when getting fluids that comes from mouth or nose” (FGD 6—Mbeya). 

Others mention that the virus that causes the COVID-19 infection has changed over time, as narrated: “COVID-19 virus changes with time and currently we have another variant that was discovered in South Africa” (IDI 2—Dar es Salaam). 

#### 3.2.2. COVID-19 Waves and Their Effects

Many participants were aware of the different waves that have occurred since 2020. However, there were variations on the number of waves, with some mentioning two, others three or five, and the majority mentioning correctly that there were four waves since the COVID-19 discovery. The participants mentioned that there were waves that hit Tanzania, and there were differences in the severity of the waves. The first and second waves were perceived to be more severe, especially by participants from urban IDIs and FGDs, while in other places, participants mentioned that wave four was the most severe as many people died. Across all the FGDs and IDIs, it was mentioned that the first wave was more serious, and people were confused and fearful as they did not know what to do, as narrated below: 

“In the first wave of COVID-19, people were confused as they didn’t know what to do…if you get a normal cough and you go to the health facility, they will label you as having the disease and they mix you with people who have it and you get it there. In the following waves, the fear was reduced as the test kits for the diseases were made available”. (FGD 1 Vaccinated women, FGD 5 Men—Kilimanjaro region)

“During the very first phase after discovering that this is a disease is in Tanzania and affect people, and the increasing death rate, there was so much fear because it seemed that everyone with this disease will only die, so there was so much fear and because people were not educated and did not have a clear understanding, honestly the first phase was a big problem”.(FGD 1 Vaccinated women—Lindi)

### 3.3. COVID-19 Risk Perception

Few of the participants mentioned being at a high risk of transmitting and getting the COVID-19 infection. A variety of reasons were mentioned to increase transmission. The reasons included the community not taking or following measures against COVID-19, traveling, work situations, as the majority must get out to work for their livelihood, and having infected family members.

“As a leader in the community I meet people with different problems, so what we do we have put a boundary in the office where our clients will stop when we listen to their problems. That is because we are in risk of being infected and infect others”.(IDI 1-Tabora)

### 3.4. Community Perception of the COVID-19 Disease

Participants had different opinions related to the COVID-19 disease. Participants who witnessed or contracted COVID-19 mentioned that it’s a very infectious disease similar to other diseases, such as smallpox, measles, and malaria, that happened in the past. Others perceived it to be a normal disease, such as the normal flu, referred to in Swahili as “*mafua*”, or a cough that can disappear for some time. 

In FGDs, some participants said the disease is not a problem in their areas because they have never seen a person with a problem. “… we only hear about it from people in other areas that someone died and things like that! So in our community to be honest we consider it as not a problem at all…”. They further explained that “Personally, I would say that this disease hasn’t entered here but other countries have really suffered…I mean like our region of Lindi…In our village…, I would say the almighty God has saved us from the disease. But in some other areas it got there…” (FGD 3- Unvaccinated women—Lindi).

Others saw it as a disease of the rich, international travelers, less energetic, and people who do not eat well, meaning that they eat light foods and do not do heavy manual work. As narrated by the participants, “The disease attacks the rich who does not eat well…only eating light food, does not do heavy manual works” (IDI 3—Mbeya; FGD 5 Men—Dar es Salaam).

Some people believe the disease does not exist, or that it is economic war or propaganda. This is depicted in the following narration: “If Corona existed in our communities, many of us would have died from it already” (FGD 3 Men—Dar es Salaam). Other communities perceived it as its ‘*Viroja*’ or ‘*Kurogwa*’ (meaning being bewitched) as explained by a traditional healer: “majority of the rural people, first of all they are not educated, poor they come to us, …they believe COVID-19 is a result of being bewitched” (IDI 4—Mbeya).

### 3.5. Barriers to the Uptake of the COVID-19 Vaccines

The majority of the participants have not received the vaccine due to various reasons and worries. Much of the worries were reported to come from official leaders, social media, and what they hear from people in the streets, called ‘*maneno ya mtaani*’ (words from the streets). Below are the sub-themes that emerged, which explain the reasons for the low uptake of the COVID-19 vaccines.

#### 3.5.1. Contradicting Statements from Top Government Leaders

It was clear in all the FGDs and IDIs that top government leaders played a big role in the low uptake of COVID-19 vaccines. All participants mentioned that the previous ruling government and technical experts during the first wave of COVID-19 had a different view on the COVID-19 disease and COVID-19 vaccines. The participants explained that some top government leaders did not believe in the vaccine and encouraged the Tanzanians to go for traditional medication and immolation “*kujifukiza*” as narrated below by participants in some of the FGDs. 

“We still remember what the top leader said that the COVID-19 vaccines are not good…mentioning that they are doing trials to African population…” (FGD 5 Men—Mbeya) and in Simiyu “…we think he was correct…” (FGD 5 Men—Simiyu).

The participants further mentioned the use of vaccines was discouraged during that time; the Tanzanians were made to believe that the COVID-19 vaccines were not good as they were meant to harm them. The participants declared that statement still lives in their hearts, and many in the community still believe it, as narrated by one participant during the IDI: “Yes you are scared and not confident after hearing such statement from the top leader. So that’s is the perception of many people in the community… Due to that some people till today they know that vaccines have a lot of effects, though some people have been vaccinated now” (IDI 3—Simiyu).

#### 3.5.2. Vaccine Rollout Preceded Education about the COVID-19 Vaccines

The majority of participants reported that they felt they were not informed well about the COVID-19 vaccines before the vaccines were rolled out. They mentioned that the idea of receiving a vaccine was very abrupt, and little information was provided to help people understand. This was contrary to the massive education provided when the COVID-19 disease was announced in the country. As reported during FGD, “…the vaccines were introduced too abruptly, there was no education about the vaccines…we were left with a lot of questions why is that…” (IDI 5—Lindi). The participants further mentioned that “the situation was quite different when the government was educating Tanzanians about the disease itself a lot of information was provided and every community members understood and were able to take precautions” (IDI 6—Dar es Salaam).

“Education given to people was not good as we still have inadequate knowledge concerning vaccine…so we still have unanswered questions like how long does the vaccine stay in the body”…“We hear about the booster, when does one need it? Does it mean the vaccine doesn’t work well? (FGD 4 Unvaccinated women—Kilimanjaro). “When they passed door to door and give us vaccines, they did not talk about repeating. Who is supposed to repeat? Eee we do not have in-depth information on this new vaccine” (FGD 2 Vaccinated women—Kilimanjaro). 

As reported across all FGDs and IDIs, sub-optimal information created hesitancy to the COVID-19 vaccine uptake “… So, in short we are still hesitating…This is because we do not have adequate education about COVID-19 vaccines that’s why most of us have not yet vaccinated” (FGD 4 Unvaccinated women—Dar es Salaam). 

#### 3.5.3. Fear/Myths towards COVID-19 Vaccines

Across all the FGDs and IDIs, participants mentioned various beliefs towards COVID-19 vaccines. The belief comes from what they have heard on social media and what is spread by people on the streets regarding vaccines. The beliefs included COVI9-19 vaccines can change genetic makeup; be a source of the COVID-19 infection; turn people into zombies; cause death; blood clots; infertility; erectile dysfunction; means to control the African population; and is a clinical trial for the African population.

“Some people said, we were joining Freemason, some say that we will have blood clots… for sure, and others say it will cause infertility for women. It brought fear that is why we are few women who have vaccinated here… Even now people are afraid to vaccinate, because you hear so many issues and views” (FGD 3 Unvaccinated women—Tabora). Another participant mentioned “I am still hesitating to be vaccinated…my fear lies to what is said in the media, at the community and the rapid development of the vaccines…” (IDI 6—Mjini Magharibi).

The vaccine was also mentioned to infect more people, as narrated, “Most have not taken it yet, like three quarters of people. Most still have misconceptions… Some say because you can still get infected with COVID-19 even after vaccination, there is no need to vaccinate” (FGD 2 Vaccinated women- Dar es Salaam).

“We have heard that if we get the vaccine we will get corona, as most people believe that CORONA is a manmade pandemic so as to reduce global population. Population reduction is one of the agenda so as to fight high population growth in some countries”. (IDI 2—Pemba; FGD 3—Kilimanjaro; FGD 2 Unvaccinated women—Lindi)

#### 3.5.4. Presence of Different Types of COVID-19 Vaccines

Across all the FGDs and IDIs, there were concerns about the different types of vaccines available in the country. The majority of the participants had a lot of questions about available vaccines, such as, “Why are we having different vaccines for the same disease?”, “Why do vaccines have different dosages some once others you have to repeat?”, “What is the effectiveness of different doses?”, and “Which is the best vaccine among all available vaccines?” The availability of different COVID-19 vaccines has made participants skeptical and hesitant towards the vaccines. The narratives below explain various concerns that the community had about COVID-19 vaccines.

“The first vaccine (JJ) was given only once; why these other ones we are supposed to get two doses?…Does it mean the first one was not working?” (FGD 3 Unvaccinated Kilimanjaro). “…they say the first vaccine which was given as a single dose was good and effective compared to the current one given twice” (FGD 3 Unvaccinated women—Kilimanjaro).

“I am confused, there are several COVID-19 waves, is it each wave has its own vaccine? (FGD 4 Men—Mbeya). These vaccines are not good, why are there many types of vaccine for a single disease? We are used to having one vaccine for one disease for example Polio. Why so many for this new disease?” (FGD 5 Men—Kilimanjaro).

#### 3.5.5. The Process of Getting the COVID-19 Vaccines

Participants had various concerns about the process of getting the COVID-19 vaccine. The participants mentioned that it has never happened in the past that if you need a vaccine, there is a form that you need to sign, and vaccine uptake has never been a choice due to its importance.

##### Signing the Consent Form

The participants mentioned that many were hesitant because they had to consent to receive the COVID-19 vaccine. This is not the case in other routine vaccines in the country, where people have to sign to the consent. 

The participants had a feeling that there was a hidden agenda that the government did not want to reveal. They were worried as to why they were requested to sign the form before being vaccinated. Additionally, in the form, there was a section stating that the government is not accountable in case of anything; it was assumed that there must be a hidden agenda, as narrated by participants “Informed consent said in case of any side effect the government will not be responsible”.…you are told if you agree to get it you commit yourself there, that is what brought too many questions to people” (FGD 4—Unvaccinated women—Kilimanjaro). It was further mentioned that “Why these people want us to fill the forms to be vaccinated? In the past we were vaccinated with other vaccines, but we didn’t fill any form, why do we have to fill the form to get COVID-19 vaccine? There must be something wrong with this vaccines” (IDI 6—Simiyu).

##### Voluntary Uptake of the COVID-19 Vaccines

Across all the IDIs and FGDs among unvaccinated and vaccinated participants, it was clear that voluntary decisions to uptake the vaccines contributed to hesitancy and low uptake of COVID-19 vaccines. The participants feel that if getting the COVID-19 vaccine is voluntary, then the vaccine is not important for their health, and they can live without being vaccinated. They further mentioned that if the vaccine was important, the government would have provided announcements that it is mandatory for everyone. 

“Making it voluntary has made it a challenge. If it is important and good for the people then it should be mandatory just like it is with children’s vaccines…”.(FGD 3 Unvaccinated women—Dar es Salaam)

Interestingly, participants mentioned that the majority of people misused the opportunity to make decisions on the COVID-19 vaccine. As mentioned during the FGDs and IDIs, it has never been a practice for the government to ask Tanzanians about issues related to vaccines. “All vaccines that were important like the smallpox the government stated that all people should be vaccinated. There was no choice because the experts recommended to the government they are important. Why is this different?” (IDI 5—Mbeya). 

“Others accept it and others do not because it is voluntary. Here people misuse their freedom hence many are not vaccinated”.(FGD 3 Unvaccinated women—Dar es Salaam)

“The response is not so good when you look at the number of people who have taken the vaccine compared to the number of Tanzanians. This may be mainly contributed to the fact that it is voluntary”.(FGD 6 men—Dar es Salaam)

#### 3.5.6. The Role of Social Media and Informal Meeting Places on Vaccine Hesitancy

All the participants were concerned that social media (Instagram, Facebook, and Twitter) and some people in the community (random people; in this context a lay person in the community that was sharing information about the COVID-19 vaccine) had created worries and are still spreading bad news about the COVID-19 vaccines. The participants mentioned that the information is spread at informal street coffee hotspots ‘*vijiwe vya kahawa*’, motorbike ‘*bodaboda* stations’, and in streets. Participants reported that the spread of misinformation on social media had amplified concerns and fears about the safety and effectiveness of the vaccines. 

“There are a lot of things being said by random people in the streets “like becoming a zombie after some years, or vaccinated people will die after two years” (FGD 4 Unvaccinated women—Dar es Salaam), “they say vaccine is bad and we are hesitant to uptake the vaccine, our worries is that we have heard the side effects will come after some years, we have also seen movie series showing the side effects to those who were vaccinated with COVID-19 vaccine…most examples does not show the positive side of the vaccine” (FGD 3 unvaccinated women—Mjini Magharibi) and “Some say it will affect ability to have children etc. Such things discourage us from going for vaccination” (FGD 3 Unvaccinated women—Kilimanjaro).

In social media, various information has also been spread, as narrated here “Social media has different opinions on this matter and this has caused the variation in people’s response. Things that are being said like infertility issues are really discouraging us from getting vaccinated” FGD 4 Unvaccinated women—Dar). “What we hear is the safety profile of the vaccine is low that’s why we have fear. We have heard of cases of death after vaccination and clotting of blood. In social media I have seen one leader who started shaking and failed to speak in public because he was vaccinated. Some of us cannot get vaccinated unless they put restrictions in shops from purchasing things like cigarettes” IDI 5 Dar.

### 3.6. Conflicts with Religious Beliefs

A few participants mentioned that taking vaccines conflicts with their religious beliefs. This was consistently reported among Christians in the Mbeya region and in some interviews in Tabora and Kilimanjaro. The participants voiced that taking vaccines means you have less trust in God.

“… Yes we have heard about COVID-19 vaccines…but we think it conflicts with our belief in trusting God. So if we vaccinate it means we don’t have hope…we don’t have trust…we leave everything to God’’.(FGD 3 Unvaccinated women—Mbeya)

“In fact, there are those who accept the vaccine and agree to be vaccinated. But there are others who are rigid, their rigidity is based on faith. They will tell you my faith does not allow me to be vaccinated. Someone will tell you we were told to practice steam inhalation. Those ones we will not vaccinate until they are told otherwise by their religious leaders”.(FGD Vaccinated women—Tabora)

## 4. Discussion

The study explored various barriers to the COVID-19 vaccine uptake, including receiving contradicting statements from top leaders, vaccine rollout preceding education, myths related to the uptake of COVID-19 vaccines, the influence of social media and random people from the community, the process of getting the vaccines, and religious beliefs.

Key findings included the lack of trust from government leaders and the lack of correct information from the technical experts that ultimately contributed to vaccine hesitancy. In our study, lack of trust was contributed by the fact that the community received contradictory statements from the top leaders and technical experts, creating confusion, mistrust, and distress among the population, which was mentioned as one of the reasons for the low uptake of the COVID-19 vaccines. Similar to our study, a lack of trust in the government’s top leaders and healthcare system has also been reported elsewhere [18,20]. It is, therefore, important for the leaders to provide consistent messages in order to increase the COVID-19 vaccine uptake and correctly advise the government on the scientific measures to take. 

In this study, participants had concerns that vaccines preceded education, and this created fear for the uptake of the vaccine. Likewise, other studies reported similar findings [18,20]. Fear of side effects and myths related to the vaccine uptake can be avoided if COVID-19 technical teams provide the community correct information about the COVID-19 virus. This may help clear the existing myth in the community. In low- and middle-income countries, the fear of side effects such as infertility, population control strategy, biological war, and micro-chip insertion was the reason for hesitancy [18,21,22]. 

Social media has an important role in transmitting health information and the uptake of certain health interventions. However, in this study, social media worsened vaccine hesitancy among community members. Social media was reported to spread information about the side effects of the vaccines. Various studies have also reported that social media has been a source of misinformation about COVID-19 vaccines hence influencing the community to not uptake the vaccines [4,5,23,24]. 

On the other hand, random people from the community played a big role in the transmission of misinformation regarding the COVID-19 vaccines. This result shows that there is a gap in information and health promotion concerning the COVID-19 vaccines in the community. It also shows that random people are very influential in transmitting information. When examining the misinformation provided by random people, the misinformation is grounded in conspiracy theories that the vaccines are meant to change genetic makeup, turn individuals into a vampire, and a means to depopulate Africa. All these conspiracies created hesitancy. Other studies have reported similar findings [24,25].

COVID-19 uptake was also linked to religious beliefs, with participants reporting that the uptake of the vaccine conflicts with their trust in God and being a mark of a beast. A review by [24,26] reported similar findings.

### Study Limitation

The study has various limitations; however, the limitations were avoided in the whole process of the study. The study used the vaccine wastage rate as a proxy for vaccine hesitancy, and this might have introduced bias, but during the designing of this study, the vaccine wastage was approximately the uptake of the vaccines. The key limitation would have arrived if there were geographical barriers to COVID-19 availability and access. However, studies both by Msuya et al. (2023) and Amour et al. (2023) showed that less than 5% of the Tanzanians reported any geographical barrier to accessing the vaccine. The study used IDIs and FGDs as a method of data collection; however, the methods introduce biases to both interviewer and participants. The expected potential biases during FGDs were dominance effects, poor group interaction, and poor facilitation skills. To minimize the biases the study used well-trained facilitators with experience in conducting FGDs. During the discussion, all participants were welcomed and encouraged to share their experiences, and all participants were identified using numbers to ensure anonymity and provide them the freedom to talk. The facilitators reminded the participants about confidentiality even during the discussion so as to aid silent participants to participate.

Another potential source of bias was a pleasing bias, where the participants share what they think the researcher wants to hear. To avoid this source of bias for IDIs and FGDs, the research team did not reveal their profession to participants and assured them that their views were vital and respected, and there were no correct or incorrect answers. In both IDIs and FGDs, the facilitators used probes, and all participants chose the place where they felt comfortable sharing their experiences. All participants were reminded about confidentiality while welcoming them freely to share their experiences.

## 5. Conclusions and Recommendations

The results of this study indicated that the community is aware of the COVID-19 infection and acknowledged having heard about the COVID-19 vaccines. Various barriers, such as fear of COVID-19 vaccines, the influence of social media and random people, conflicting information from the government, and inadequate education about COVID-19 vaccines, led to low vaccine uptake. The findings of this study imply that the community knowledge of vaccines is still low, and this has influenced the uptake of COVID-19 vaccines. There is a need for interventions that focus on increasing community knowledge about COVID-19 vaccines and addressing myths about the vaccines that are needed. This can be possible through the use of technical experts and involving trusted people in the community that can influence the uptake. Technical experts need to provide facts about the vaccines and provide accurate information through trusted sources such as health care providers. At the community level, the use of trusted people such as religious or community leaders can help spread the correct information about vaccines; however, the technical experts need to train these people and provide them with fact sheets about the vaccines. 

## Figures and Tables

**Table 1 vaccines-11-01366-t001:** Background characteristics of participants in FGDs (N = 369).

Variable	*n* (%)
Regions	
Mbeya	50 (13.5)
Kilimanjaro	47 (12.7)
Simiyu	41 (11.1)
Lindi	55 (14.9)
Tabora	60 (16.2)
Dar es salaam	39 (10.6)
Mjini Magharibi	31 (8.4)
Kaskazini Pemba	46 (12.4)
Sex	
Male	116 (31.4)
Female	253 (68.6)
Age range in years (min–max)	(18–94)
Mean age (SD) in years	40 (13.2)
Marital Status	
Single	71 (19.2)
Married	239 (64.7)
Separated	22 (5.9)
Widow	37 (10.0)
Educational level	
Incomplete primary school	31 (8.4)
Primary school	208 (56.4)
Secondary school	106 (28.7)
Diploma/certificates	12 (3.3)
Higher education	12 (3.2)
Occupation	
Employed	28 (7.6)
Self employed	188 (50.9)
Not employed	105 (28.5)
Farming	48 (13.0)
Vaccination status (females *n* = 253)	
Vaccinated	127 (49.8)
Unvaccinated	126 (50.2)

**Table 2 vaccines-11-01366-t002:** Background characteristics of participants in IDIs.

Variable	*n* (%)
Sex	
Male	26 (81.3)
Female	6 (18.7)
Age range in years (min–max)	22–74
Mean age (SD) in years	45.4 (13.1)
Marital Status	
Single	3 (9.4)
Married	29 (90.6)
Educational level	
Incomplete primary school	2 (8.3)
Primary school	6 (25.0)
Secondary school	7 (29.2)
Diploma	3 (12.5)
Higher education	6 (25.0)
Occupation	
Employed	15 (46.9)
Not employed	12 (37.5)
Self employed	5 (15.6)

## Data Availability

Data is available upon reasonable request due to the potential sensitivity of the data.

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
