# Peer review of "A Qualitative Study on Barriers to COVID-19 Vaccine Uptake among Community Members in Tanzania"

_vaccines, 2023, doi:10.3390/vaccines11081366_

Round 1
Reviewer 1 Report
The authors conducted a qualitative study on factors affecting Covid19 vaccine uptake through group discussions and individual interviews in different geographic regions of Tanzania. The results identified several factors many of which have previously been related to vaccine hesitancy in the context of Covid-19 vaccination.
Comments:
1. The authors conducted their research in urban and rural areas of Tanzania. Did they observe differences between these areas?
2. The manuscript includes terms in the local language (presumably Swahili). That is fine, but the authors should add in parentheses the English translation. E.g. lines 205 and 335.
3. The term “vaccine preceded education” (lines 27, 374) is not clear. The authors mean that there was insufficient information provided about the vaccines before the vaccinations were rolled out. This should be explained better in the text.
4. Line 419-420. “Ethical approval was also requested…”. Did the authors receive approval?
5. Several of the references (e.g.3 – 5) are incomplete as they lack the name of the journal.
6. The manuscript has several grammatical and typographical errors, e.g. lines 41 (“these included” should be the start of a new sentence), line 58 (“There are five COVID-19 vaccines that are available…” change to Five Covid-19 vaccines are available”), line 65 (“low” should be “lower”), line 166 (“aaah”), and lines 378-379 (incomplete sentences).
See item 6. above
Author Response
Please see the attchment

Reviewer 2 Report
Abstract:
The abstract does not provide any information on the implications of the study's findings or how they can be used to improve COVID-19 vaccine uptake in Tanzania. Therefore, it would be beneficial for the authors to discuss this aspects at the end of the manuscript.
Methodology:
1. Study design: The authors used a mixed-methods study design; however, the manuscript only focuses on qualitative data. It would be useful to know more about the quantitative aspect of the study, including the sampling method, sample size calculation, and statistical analysis.
2. Sampling method: The authors purposively selected the study regions based on the vaccine wastage rate of September 2021. This approach may introduce bias into the study, as it assumes that vaccine wastage rate is a proxy for vaccine hesitancy. The authors need to provide a rationale for using this approach and its potential limitations.
3. Sample size: The manuscript does not provide a clear justification for the sample size of the study. The authors need to explain how they arrived at the number of participants included in the study, especially given the large number of regions and districts involved.
4. Participant selection: The authors used a convenience sampling method to select participants for the in-depth interviews and focus group discussions. This approach may not be representative of the broader population and may bias the results of the study. The authors need to explain how they addressed this limitation.
5. Data collection: The authors collected data using in-depth interviews and focus group discussions. While these methods can provide rich and detailed data, they are also subject to interviewer and participant bias. The authors need to explain how they addressed these potential sources of bias.
6. Data analysis: The authors used thematic framework analysis to summarize the data. While this is a widely used method for qualitative data analysis, the manuscript does not provide details on how the authors arrived at their final themes and sub-themes. The authors need to provide a detailed description of their data analysis process.
Results:
The authors report that the FGDs were conducted with women who were vaccinated, women who were not vaccinated, and men aged 18 years and above. However, they do not provide any information on why these specific groups were chosen or whether there were any differences in the responses between the groups. It would have been helpful if the authors had provided a detailed analysis of the data by group, and if they had explored whether there were any differences in the factors affecting vaccine uptake among the different groups.
The authors also report that coding and thematic framework analysis were used to summarize the data. However, they do not provide any information on how the coding was done or whether inter-coder reliability was assessed. It would have been helpful if the authors had provided some examples of the codes that were used and how they were derived from the data. The lack of details on the analysis methods used in the study makes it difficult to assess the validity and reliability of the findings.
Discussion:
Firstly, the authors have discussed the role of social media in spreading misinformation about COVID-19 vaccines, but it would have been helpful to know more about the specific types of messages being circulated on social media that were contributing to vaccine hesitancy. Additionally, the authors could have discussed potential strategies for addressing misinformation on social media, such as fact-checking and providing accurate information through trusted sources.
Secondly, the discussion mentions the influence of random people in spreading misinformation about the vaccines, but it is not clear how the study participants were defining "random people." It would be useful to have more information on who these individuals were and what specific information they were sharing that was contributing to vaccine hesitancy.
Finally, while the authors have provided some recommendations for interventions to increase vaccine uptake, such as involving trusted community members and technical experts in education efforts, it would have been helpful to have more specific details on how these interventions could be implemented and what resources might be required. Overall, the discussion provides a good overview of the study's findings, but there is room for further elaboration and clarification on certain points.
Round 2
Reviewer 2 Report
As the introduction is also very important part of the manuscript, therefore it should be more appropriate,
There are a few areas where some improvements could be made.
The inclusion of previous studies and factors contributing to vaccine hesitancy strengthens the foundation for the research. However, it would be beneficial to provide specific examples or references to support the mentioned factors such as religion and culture, perceived risks and benefits, and healthcare provider recommendations. This would enhance the credibility of the discussion and demonstrate the existing literature on the subject.
The introduction effectively introduces the COVID-19 pandemic and the importance of vaccination in reducing infection rates, hospitalizations, and deaths. It highlights the disparity in COVID-19 vaccine uptake in Sub-Saharan Africa, specifically in Tanzania. The inclusion of specific statistics and data sources would further support these claims and provide a clearer picture of the situation.
The challenges faced by Tanzania in introducing COVID-19 vaccines and the subsequent vaccination campaign are well outlined. However, additional details regarding the political and leadership choices that led to the delay in vaccine introduction would provide a more comprehensive understanding of the context and potential influences on vaccine hesitancy.
To further enhance the discussion, I recommend referencing the experiences of other developing countries in vaccine introduction and development, which can offer valuable insights into addressing vaccine hesitancy. Furthermore, including references to recently published articles (e.g. https://www.mdpi.com/2076-393X/11/3/607) in the introduction would provide a clearer context and additional support for the study.
